# ZrB_2_/SiCN Thin-Film Strain Gauges for In-Situ Strain Detection of Hot Components

**DOI:** 10.3390/mi13091467

**Published:** 2022-09-04

**Authors:** Fan Lin, Xiaochuan Pan, Chao Wu, Yingjun Zeng, Guochun Chen, Qinnan Chen, Daoheng Sun, Zhenyin Hai

**Affiliations:** Department of Mechanical and Electrical Engineering, School of Aerospace Engineering, Xiamen University, Xiamen 361005, China

**Keywords:** thin film strain gauge, direct ink writing, polymer-derived ceramics, conductive composites, high temperature

## Abstract

The in-situ strain/stress detection of hot components in harsh environments remains a challenging task. In this study, ZrB_2_/SiCN thin-film strain gauges were fabricated on alumina substrates by direct writing. The effects of ZrB_2_ content on the electrical conductivity and strain sensitivity of ZrB_2_/SiCN composites were investigated, and based on these, thin film strain gauges with high electrical conductivity (1.71 S/cm) and a gauge factor of 4.8 were prepared. ZrB_2_/SiCN thin-film strain gauges exhibit excellent static, cyclic strain responses and resistance stability at room temperature. In order to verify the high temperature performance of the ZrB_2_/SiCN thin-film strain gauges, the temperature-resistance characteristic curves test, high temperature resistance stability test and cyclic strain test were conducted from 25 °C to 600 °C. ZrB_2_/SiCN thin-film strain gauges exhibit good resistance repeatability and stability, and highly sensitive strain response, from 25 °C to 600 °C. Therefore, ZrB_2_/SiCN thin-film strain gauges provide an effective approach for the measurement of in-situ strain of hot components in harsh environments.

## 1. Introduction

Thin film strain gauges (TFSGs) are widely used for in-situ strain detection of various components and structures in the aerospace, transportation and automobile industries, civil engineering and even the medical field due to their advantages of non-interference, small size, fast response and in-situ integration [1,2,3,4,5,6,7]. TFSGs are mainly fabricated by depositing alloy/metal films such as NiCr, PdCr and TaN-Cu on the surface of components [1,8,9]. An effective TFSG must exhibit a number of appropriate properties (e.g., adequate operating range, reasonable conductivity, lack of frequency dependence), probably the most important property being the sensitivity or gauge factor (GF) [10]. However, the gauge factor (GF) of traditional metal/alloy iso piezoresistive materials is about two, resulting in low sensitivity and difficulty in detecting tiny strains.

To solve this issue, many researchers have turned to conductive composites. Conductive composites contain an insulating matrix and conductive nanoparticles dispersed therein, and the conduction mechanism and the strain sensitivity are primarily dominated by the tunneling effect [10,11]. Flexible sensors based on conductive polymer composites have stretchability and strain factors that far surpass foil strain gauges, and are widely used in wearable devices, electronic skins and human motion detection, etc. [12,13,14,15,16,17]. However, most TFSGs based on conductive composites are limited to room temperature. They are not thermally stable at high temperatures.

To construct thermally stable conductive composites with a highly sensitive piezoresistive response, the high-temperature thermal stability of the insulating matrix and the conductive phase is the first design principle. Compared to metallic and polymeric materials, most ceramics, such as polymer-derived ceramics (PDCs) pyrolyzed at high temperature are electrically insulating and thermally stable at high temperatures [18]. Existing PDCs high temperature sensors such as temperature sensors and pressure sensors are still dominated by discrete bulk devices [19,20,21]. Compared with larger discrete devices, in-situ integrated TFSGs with a thickness of micrometers exhibit reproducible faster response [22]. However, the huge volume shrinkage during the pyrolysis of PDCs will cause stress mismatch at the interface of the film, leading to cracking or peeling off [23]. The use of particle fillers such as SiC, ZrO_2_ and TiB_2_ can not only reduce the shrinkage of the PDCs during the pyrolysis process, but also modify the properties of PDCs, such as electrical properties and mechanical properties [24]. Boride ceramics are excellent conductors of electricity with good mechanical properties, and oxidation resistance [25,26]. This makes boride ceramics promising for high temperature sensors and electrical functional devices [2,27].

In this study, PDC-SiCN was used as the insulating matrix, and the dispersed ZrB_2_ conductive particles acted as the conductive phase. TFSG based on ceramic conductive composites was fabricated by the direct ink writing (DIW) technique based on the Weissenberg effect. Herein, the morphologies and microstructure of the ZrB_2_/SiCN TFSGs were characterized. The effects of ZrB_2_ content amount on the electrical conductivity and GF of ZrB_2_/SiCN films were investigated. The piezoresistive response of ZrB_2_/SiCN TFSGs at room temperature was tested. Ultimately, high temperature performance of the ZrB_2_/SiCN TFSG was investigated from 25 °C to 600 °C.

## 2. Materials and Methods

### 2.1. Materials and Preparation Process

As shown in Figure 1a, commercially available PSN2 (Chinese Academy of Sciences, China) filled with ZrB_2_ nanopowder (average diameter: 50 nm, Shanghai Chaowei Nano Technology Co., Ltd., Shanghai, China) was utilized as printing ink. The filling weight percent of ZrB_2_ nanopowder is 40~60 wt%. The ZrB_2_ nanopowder were uniformly dispersed in PSN2 by magnetic stirring for more than 2 h. Briefly, as shown in Figure 1b, the ink was printed by a Weissenberg-based DIW platform, which consisted of three key components: an x–y high-precision moving platform, a homemade printing setup including a printing head and a charged–coupled device camera. The printing head consists of micron tube and microneedle. The solution is quickly transported to the printing needle through the micron tube under the high-speed rotation of the microneedles. Then, the prepared thin-film strain grids were pyrolyzed in a tube furnace under nitrogen atmosphere (−0.1 MPa is evacuated before introducing nitrogen) at 800 °C for 4 h (heating rate 2 °C/min and cooling rate 3 °C/min). Finally, Ag paste was used to prepare solder joints to connect the thin-film strain grids and the Pt leads.

### 2.2. Experiment Setup

Strain grid thicknesses were determined by a profilometer (Dektak XT, BRUKER, Billerica, MA, USA). SEM (SUPRA55 SAPPHIRE, CARL ZEISS, Oberkochen, Battenburg, Germany) coupled with EDS was used to characterize the morphology of the obtained samples. High temperature furnace (GSL-1700X, HF. Kejing, Hefei, China) was used to pyrolysis and high temperature furnace (OTF-1200X, HF. Kejing, Hefei, China) was used to test in high temperature.

A cantilever beam arrangement was used to investigate the strain response behavior of the ZrB_2_/SiCN TFSGs, as shown in Figure 2a. One end of the beam was clamped, and the sensor was subjected to strain by applying displacement at the free end of the cantilever [2]. The corresponding resistance changes of the TFSG were recorded using data acquisition equipment. Calculate the strain at the location of the strain gauge according to the following Equation (1) [28]:(1)$$\varepsilon =\frac{3yhx}{2l^{3}}$$
where ε is the strain at the location of the TFSGs, *y* is the deflection at the free end, *l* is the length of the cantilever beam, *x* is the distance from center of strain gauge to the point of application of load and *h* is the thickness of the beam. The indicator for strain sensitivity of the strain gauge is defined as:(2)$$\mathrm{GF}=\frac{∆R/R_{0}}{\varepsilon }$$
where Δ*R* is the change of TFSG resistance when strain ε is applied and *R*_0_ is the initial resistance of TFSG. The piezoresistive response of TFSG at high temperatures was done in a high temperature furnace. The tube furnace is heated to 600 °C at 12 °C/min. In the meantime, the stepper motor is applied strain to the free end of the cantilever beam to obtain the strain response at high temperatures.

The high temperature resistance test system of the strain gauge is shown in Figure 2b, which consists of a tube furnace and a standard k-type thermocouple. The unit of TCR is ppm/°C and is used to express the relationship between the resistance of the strain gauge and the temperature. TCR can be calculated by the following Equation (3) [29]:(3)$$\mathrm{TCR}=\frac{dR}{RdT}\times 10^{6}$$

## 3. Results

### 3.1. Microstructural Characterisation of ZrB_2_/SiCN TFSG

The fabricated ZrB_2_/SiCN TFSGs on Al_2_O_3_ substrate are shown in Figure 3a. The length and width of ZrB_2_/SiCN TFSGs are 7 mm and 5 mm, respectively. Its line width and thickness were determined by the profilometer, and were 600 μm and 15 μm, respectively. Porosity, cracks and inhomogeneity are the main factors affecting the electrical conductivity and thermal stability of TFSG. The low-magnification SEM image of ZrB_2_/SiCN TFSGs is shown in Figure 3c. There are no obvious cracks on the surface of ZrB_2_/SiCN TFSGs. The high-magnification SEM image of ZrB_2_/SiCN TFSGs in Figure 3d shows a dense and crack-free surface. The SEM cross-sectional image presented in Figure 3e shows that the interface is clearly visible, and the sensitive grid is tightly bonded to the substrate without an obvious gap.

### 3.2. Piezoresistive Response of ZrB_2_/SiCN TFSG

The electrical conductivities of the printed ZrB_2_(40 wt%)/SiCN, ZrB_2_(50 wt%)/SiCN, ZrB_2_(60 wt%)/SiCN strained grids are 0.036 S/cm, 0.077 S/cm and 1.71 S/cm, respectively. With the increase of ZrB_2_ nanopowder filling, the electrical conductivity of the strain grids increases significantly, which is related to the conductive network composed of ZrB_2_ in ZrB_2_/SiCN composites. To investigate the piezoresistive behavior of ZrB_2_/SiCN TFSGs in detail, their strain responses were tested using the deflection method at room temperature. The strain responses of the ZrB_2_(40 wt%)/SiCN TFSG at room temperature are shown in Figure 4a–d. Figure 4a shows the static strain response of the ZrB_2_(40 wt%)/SiCN TFSG. Strain was applied sequentially in 100 με increments, and the change in resistance was consistent with the strain applied to the sensor as time progresses. The ZrB_2_(40 wt%)/SiCN TFSG exhibits a good response that stepwise applied strain leads to a distinguishable, recoverable step change in the resistance of the TFSG. The ZrB_2_(40 wt%)/SiCN TFSG exhibits a positive GF, that is, the resistance increases with increasing positive strain and decreases with increasing negative strain. Figure 4b shows the strain responses of ZrB_2_(40 wt%)/SiCN TFSG under different strain amounts, where strains of 100 με, 200 με, 300 με, 400 με, and 500 με were sequentially applied to the TFSG at a constant strain rate of 100 με/s. Figure 4c,d show the strain response of 500 με at different strain rates (20 με/s, 50 με/s, 100 με/s, 200 με/s, and 400 με/s) and the cyclic strain response with a period of 4 s, respectively. The applied strains were all 500 με. Consistent changes in relative resistance indicate that the ZrB_2_(40 wt%)/SiCN TFSG has a stable and strain-rate independent strain response.

The strain responses of ZrB_2_(50 wt%)/SiCN TFSG at room temperature are shown in Figure 5a–d. Similar to ZrB_2_(40 wt%)/SiCN TFSG, ZrB_2_(50 wt%)/SiCN TFSG exhibits stable, distinguishable strain responses.

The strain responses of ZrB_2_(60 wt%)/SiCN TFSG at room temperature are shown in Figure 6a–d. Compared with ZrB_2_(40 wt%)/SiCN and ZrB_2_(50 wt%)/SiCN TFSGs, the strain signal of ZrB_2_(60 wt%)/SiCN TFSG is more obvious.

The GFs of the ZrB_2_/SiCN TFSGs were calculated according to Equation (2). The GFs of ZrB_2_(40 wt%)/SiCN, ZrB_2_(50 wt%)/SiCN and ZrB_2_(60 wt%)/SiCN at room temperature are 3.4, 3.3 and 4.8, respectively (see Figure 7). Comparing the GF of TFSG with different ZrB_2_ filling amount, the GF of ZrB_2_(60 wt%)/SiCN is the highest, which is mainly owing to the change in resistivity by the concentration of ZrB_2_ conductive phase, which leads to the increase of the effect of piezoresistive response. Guenter Schultes et al. fabricated boride TFSG on a Al_2_O_3_ substrate by DC magnetron sputtering and obtained 0.7 GF [26]. In contrast, the GF of TFSG based on ZrB_2_/SiCN conductive ceramic composite is several times higher than that of single boride TFSG.

### 3.3. High Temperature Performance of the ZrB_2_/SiCN TFSG

Since ZrB_2_(60 wt%)/SiCN TFSG exhibited higher conductivity and strain sensitivity, its high temperature performance was tested. In practical applications, the consistency of the temperature-resistance characteristics during the cycle temperature and high temperature resistance stability of the TFSG are very important, and it reflects the stability and oxidation resistance of the TFSG at high temperatures. In order to study the repeatability of the temperature resistance of ZrB_2_(60 wt%)/SiCN TFSG, we tested the temperature-resistance characteristic curves of two times of heating and cooling (Figure 8a). ZrB_2_(60 wt%)/SiCN TFSG exhibited a negative temperature coefficient of resistance of −428 ppm/°C and good repeatability in the range of 25–600 °C. Figure 8b shows the resistance change curves of ZrB_2_(60 wt%)/SiCN TFSG at 600 °C for 2 h. ZrB_2_(60 wt%)/SiCN TFSG exhibited excellent resistance stability and antioxidant qualities at 600 °C, and resistance increased by 1.3% after two hours of oxidation. The good repeatability, stability and consistency of the resistance of ZrB_2_(60 wt%)/SiCN TFSG are attributed to the oxide layer formed on the surface of the film, which prevents the further diffusion of oxygen [30].

To evaluate the strain response of ZrB_2_(60 wt%)/SiCN TFSG at high temperatures, cyclic strain tests were carried out from room temperature to 600 °C, as shown in Figure 9a. The curves of the cyclic strain response were intercepted at 400 °C, 500 °C and 600 °C, respectively, as shown in the Figure 9b–d. ZrB_2_(60 wt%)/SiCN TFSG exhibits good repeatability and stability resistance at high temperatures. Although the overall resistance decreases with increasing temperature due to the temperature-resistance effect, the pulse signal caused by the cyclic strain is clearly visible. The above high-temperature test results show that ZrB_2_(60 wt%)/SiCN TFSG has good resistance stability and highly sensitive strain response in the temperature range from room temperature to 600 °C, and has potential application in the field of hot component strain monitoring/sensing.

## 4. Discussion

ZrB_2_/SiCN TFSGs were fabricated on alumina substrates by DIW of the Weissenberg effect. The used DIW process enabled thin-film patterning and in situ strain/stress sensing of high-temperature components. The ZrB_2_/SiCN TFSGs were characterized to determine the structural dimensions, surface topography and cross-sectional structure by SEM. The piezoresistive behavior of ZrB_2_/SiCN TFSGs at room temperature was investigated by the deflection method. ZrB_2_/SiCN TDSGs exhibited excellent strain responses and resistance stability at room temperature. The effects of ZrB_2_ content on the electrical conductivity and strain sensitivity of ZrB_2_/SiCN composites were investigated. Finally, ZrB_2_(60 wt%)/SiCN film with high conductivity (1.71 S/cm) and GF of 4.8 was used as the sensitive material for high-temperature thin-film strain gauges. The temperature-resistance characteristic curves of ZrB_2_(60 wt%)/SiCN TFSGs were tested, and the TFSGs exhibited a negative temperature coefficient of resistance of −428 ppm/°C and good repeatability in the range of 25–600 °C. The resistance change curves of ZrB_2_(60 wt%)/SiCN TFSGs were tested at 600 °C. The TFSGs have good resistance stability with resistance increasing by 1.3% after two hours of oxidation at 600 °C. Finally, the strain response verification was conducted from 25 °C to 600 °C. ZrB_2_(60 wt%)/SiCN TFSG has a highly sensitive strain response from 25 °C to 600 °C. Therefore, ZrB_2_/SiCN TFSGs based on the Weissenberg DIW can be applied to micro-strain detection from room temperature to 600 °C. Further research is underway to improve the antioxidant nature of ZrB_2_/SiCN TFSGs for applying to higher temperatures.

## Figures and Tables

**Figure 1 micromachines-13-01467-f001:**
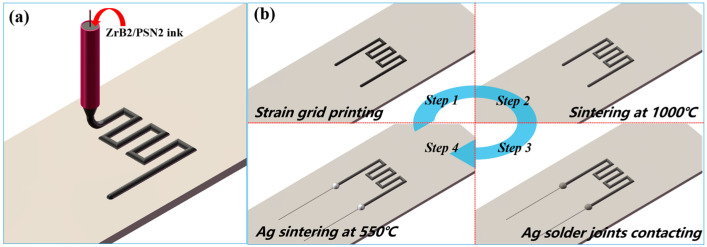
(**a**) Direct ink writing patterning of the strain gauge; (**b**) Schematic of the ZrB_2_/SiCN thin-film strain gauge fabrication process.

**Figure 2 micromachines-13-01467-f002:**
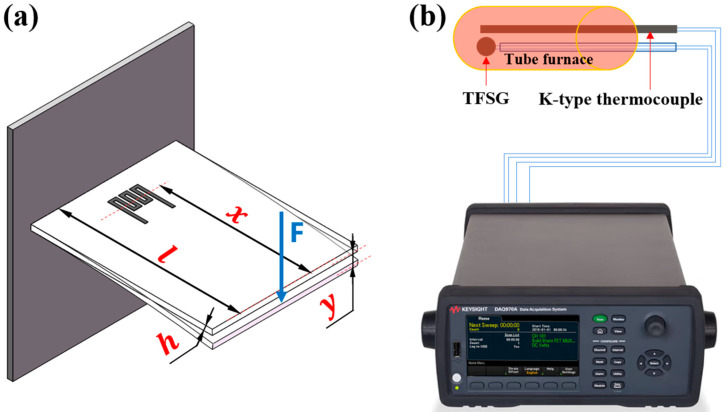
Schematic of experiment setups for (**a**) strain response testing and (**b**) temperature resistance characteristics testing.

**Figure 3 micromachines-13-01467-f003:**
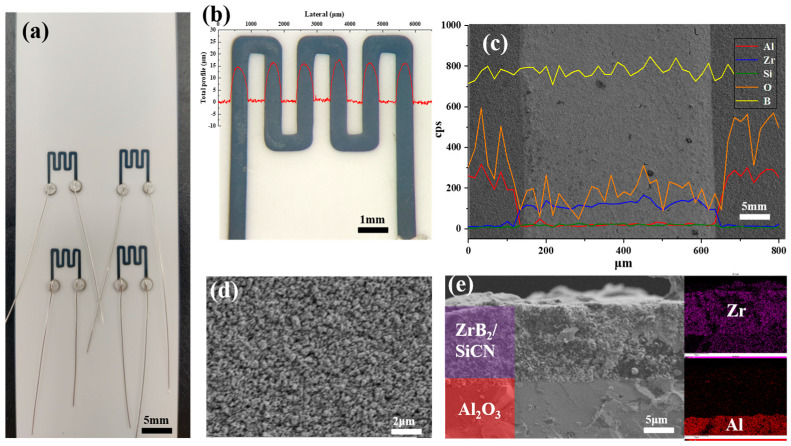
(**a**) Samples of the thin film strain gauge; (**b**) Thickness and width of strain grids; (**c**) Low-magnification SEM image and EDS analysis of TFSGs; (**d**) High-magnification SEM image of TFSGs; (**e**) The cross-sectional SEM images and EDS analysis of TFSGs.

**Figure 4 micromachines-13-01467-f004:**
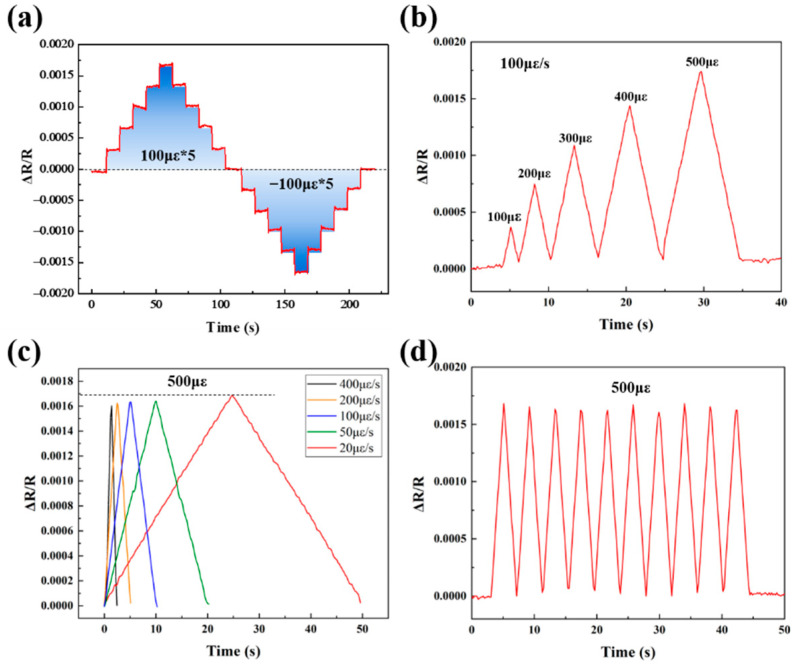
Piezoresistive response of the ZrB_2_(40 wt%)/SiCN TFSG: (**a**) The static response; (**b**) The response of different strain quantities at a constant strain rate (100 με/s); (**c**) The response at different strain rates; (**d**) The cyclic response.

**Figure 5 micromachines-13-01467-f005:**
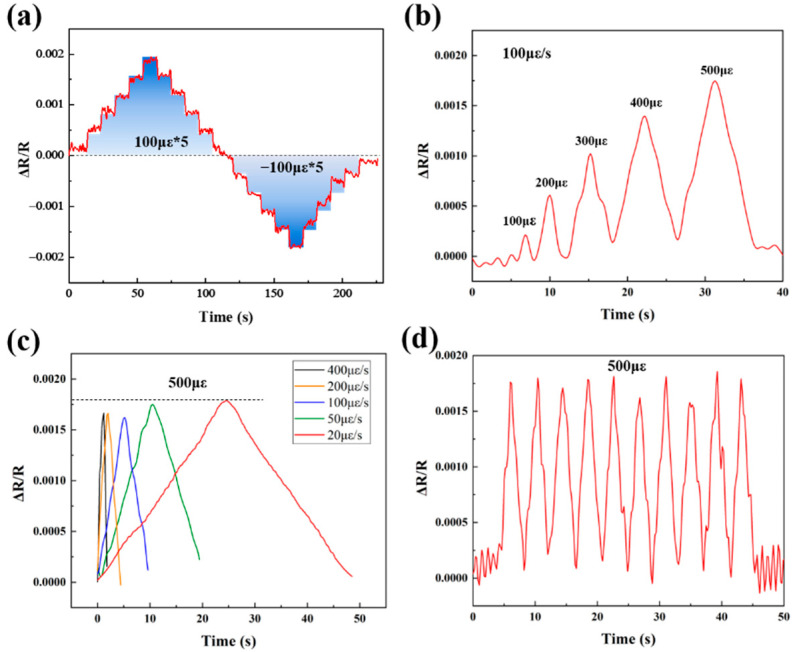
Piezoresistive response of the ZrB_2_(50 wt%)/SiCN TFSG: (**a**) The static response; (**b**) The response of different strain quantities at a constant strain rate (100 με/s); (**c**) The response at different strain rates; (**d**) The cyclic response.

**Figure 6 micromachines-13-01467-f006:**
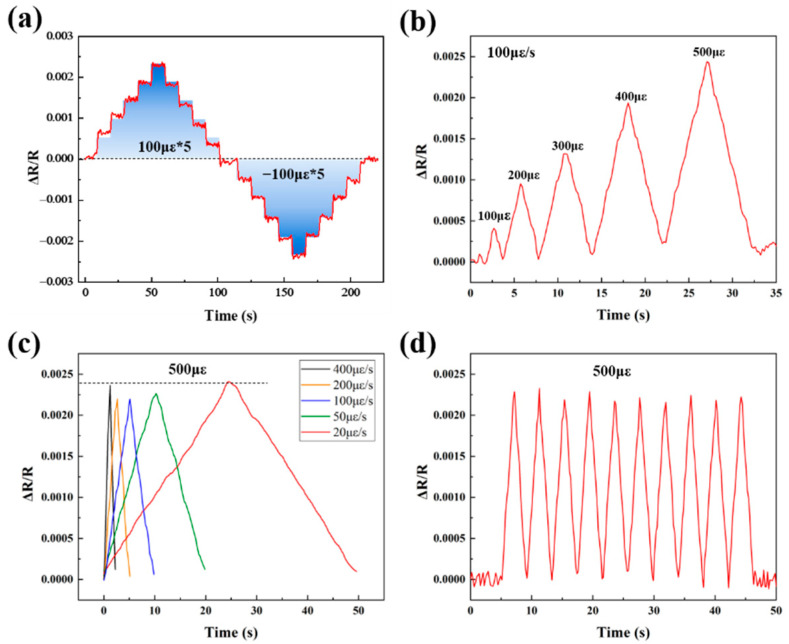
Piezoresistive response of the ZrB_2_(60 wt%)/SiCN TFSG: (**a**) The static response; (**b**) The response of different strain quantities at a constant strain rate (100 με/s); (**c**) The response at different strain rates; (**d**) The cyclic response.

**Figure 7 micromachines-13-01467-f007:**
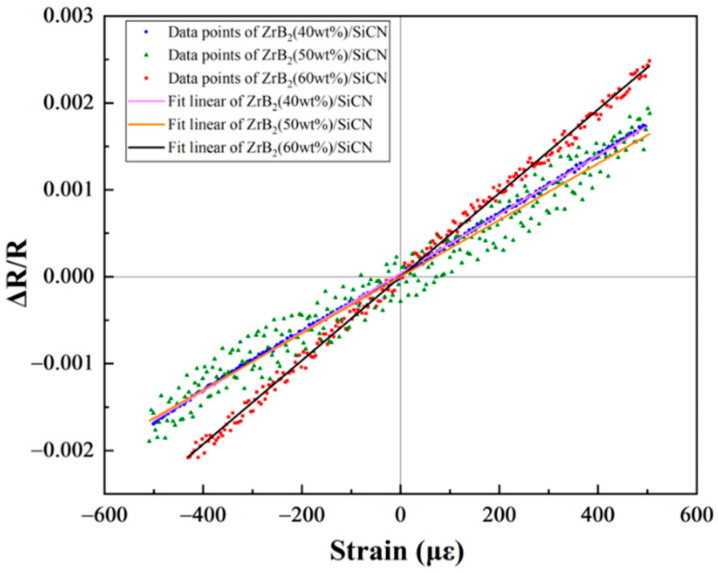
The GFs of the ZrB_2_/SiCN TFSGs.

**Figure 8 micromachines-13-01467-f008:**
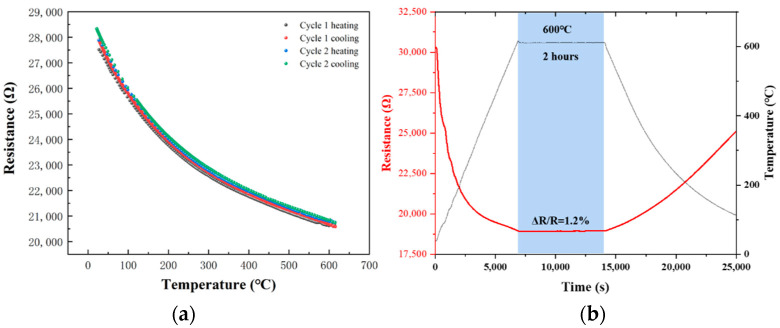
(**a**) R–T curves of ZrB_2_(60 wt%)/SiCN TFSG from room temperature to 600 °C; (**b**) Resistance change curves of ZrB_2_(60 wt%)/SiCN TFSG at 600 °C for 2 h.

**Figure 9 micromachines-13-01467-f009:**
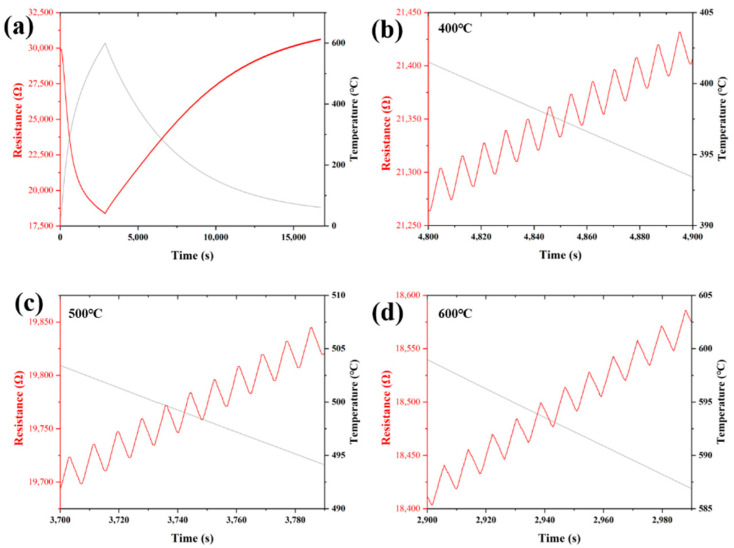
(**a**)The cyclic response from 25 °C to 600 °C; (**b**) The cyclic response at 400 °C; (**c**) The cyclic response at 500 °C; (**d**) The cyclic response at 600 °C.

## Data Availability

Not applicable.

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
