# Peer review of "ZrB2/SiCN Thin-Film Strain Gauges for In-Situ Strain Detection of Hot Components"

_micromachines, 2022, doi:10.3390/mi13091467_

Round 1

Reviewer 1 Report

In this paper, the authors use the direct ink writing method to fabricate ZrB2/SiCN thin-film strain gauges. They presented exciting research on the effects of ZrB2 concentration on their performance as well as the high-temperature performance. The comments are as follows:

1.      Figure 3, (c-d), how to conduct the TEM characterization of the thin film strain gauge? Besides, the scale bar in (d) is too large to be called a “High-magnification TEM image”.

2. In the figure caption of Fig. 6, the weight percentage of ZrB2 is supposed to be 60% according to the context.

3.      What about the higher filling amount (>60 wt%) or a lower amount (<40 wt%) of ZrB2? What is the effect of ZrB2 concentration on the piezoresistive response?

4.      In ref. 2, the authors also studied the TiB2/SiCN thin film strain gauges, compared with TiB2, what is the advantage of ZrB2? And the performance should also be compared of the two types of TFSGs.

5.      The scale bar in Figure 3 is not clear.

Reviewer 2 Report

The authors fabricated the ZrB2/SiCN thin-film strain gauges on alumina substrates by direct writing. High temperature performance of the ZrB2/SiCN thin-film strain gauges was studied from 25℃ to 600℃. However, the analysis process is not sufficient to support the interesting works. My detailed comments are listed below.  

1.     The differences and innovations between this work and the existing work of PDC SiCN should be discussed in the introduction.

2.     The details of the direct ink writing technique should be given, and the parameters of the different inks also should be described in detail so that other researchers can repeat the experiment.

3.     What is the inert atmosphere, argon? Nitrogen? How much pressure is evacuated before introducing inert gas, and the experimental details should be given. The equipment used for high-temperature sintering should be introduced.

4.     What is the size of the sensor (length and width)?

5.     How is the binding force between the sensor and the substrate? and the binding force test data should be given.

6.     Why use cantilever beam test instead of constant strain beam substrate? How to determine the value of X in equation (1)? Because the position of the TFSG shows a gradient change in the axial direction of the cantilever beam.

7.     In 3.1, the Fig.3c should be SEM image instead of TEM. And the scale bar should be clearer in Fig.3. Which parameter is the TFSG in Figure 3? ZrB2(40wt%)? ZrB2(50wt%)? or ZrB2(60wt%)? Is there any differences between the SEM images of different TFSGs? Why not give the SEM images of all TFSGs. In addition, What intention of the EDS curve on the SEM image?

8.     In 3.2, the authors claimed that the dynamic strain response at a frequency of 0.25Hz. Actually, the test frequency is too low to prove any valuable results. The author should make it clear what dynamic strain testing is. In addition, the loading curve should be given in the figures to evaluate the piezoresistive response of the TFSG.

9.     In 3.2, the authors claimed that the ZrB2(60wt%)/SiCN has good resistance stability. Authors should calculate the resistance drift rate to quantitatively describe its stability. The reasons for its stability should be analyzed more clearly.

Round 2

Reviewer 2 Report

The piezoresistive response test with a period of 4 seconds (0.25HZ) cannot be called dynamic piezoresistive response test. If the authors insist that it is dynamic test, please provide test results with higher frequency (e.g. 100Hz). Otherwise it cannot be called dynamic test. Because it will mislead other researchers in this field.

Author Response

All the ‘’dynamic strain‘’ are removed or replaced with “cyclic strain“,thank you for your correction.